# OCR Applied for Identification of Vehicles with Irregular Documentation Using IoT

**Luiz Alfonso Glasenapp** [1], **Aurélio Faustino Hoppe** [1], **Miguel Alexandre Wisintainer** [1], **Andreza Sartori** [1] **and Stefano Frizzo Stefenon** [2,3,*]

[1]   Department of Information Systems and Computing, Regional University of Blumenau,
    Rua Antônio da Veiga 140, Blumenau 89030-903, Brazil
[2]   Digital Industry Center, Fondazione Bruno Kessler, Via Sommarive 18, 38123 Trento, Italy
[3]   Department of Mathematics, Computer Science and Physics, University of Udine, Via delle Scienze 206,
    33100 Udine, Italy
[*]   Correspondence: sfrizzostefenon@fbk.eu

**Abstract:**   Given the lack of investments in surveillance in remote places, this paper presents a prototype that identifies vehicles in irregular conditions, notifying a group of people, such as a network of neighbors, through a low-cost embedded system based on the Internet of things (IoT). The developed prototype allows the visualization of the location, date and time of the event, and vehicle information such as license plate, make, model, color, city, state, passenger capacity and restrictions. It also offers a responsive interface in two languages: Portuguese and English. The proposed device addresses technical concepts pertinent to image processing such as binarization, analysis of possible characters on the plate, plate border location, perspective transformation, character segmentation, optical character recognition (OCR) and post-processing. The embedded system is based on a Raspberry having support to GPS, solar panels, communication via 3G modem, wi-fi, camera and motion sensors. Tests were performed regarding the vehicle's positioning and the percentage of assertiveness in image processing, where the vehicles are at different angles, speeds and distances. The prototype can be a viable alternative because the results were satisfactory concerning the recognition of the license plates, mobility and autonomy.

**Keywords:** surveillance; irregular vehicles; license plate recognition; OCR; IoT; embedded system

## 1. Introduction

Property security is a set of private security practices and measures ensuring the integrity of the property and the physical safety of people. There are several practices that contribute to property security, especially prevention, control, detection and intervention [1]. One way to assist in prevention is the use of electronic security that provides protection through systematized and integrated equipment, such as alarms and monitoring cameras.

The use of monitoring cameras is one of the most efficient means for the protection and control of assets and personal safety. Through them, it is possible to see and record images of vulnerable or risky places, located in residential, corporate and public environments. In addition, they also inhibit the actions of intruders and malicious people, facilitate the work of the police and private surveillance and are easily integrated into alarm systems with access to images via the Internet [2].

Prevention can also occur through collaborative security which is a set of actions to unite society in favor of common surveillance measures. It is possible to increase the level of security through electronic equipment, such as cameras, which help in remote monitoring, enabling a group of people to perform the triggering of a responsible body or person in case of suspicious activities such as a robbery attempt [3].

The proposal of an Internet of things (IoT) system [4] to build a social network for neighbors consists of a platform where neighbors monitored certain regions through

cameras connected to the cloud [5]. The captured images can be checked at any time and from any device, only requiring a user and password.

The main objectives of this research are: (i) integrate the Raspberry Pi 3 with a Global Positioning System (GPS) module, a camera and an OpenCV image processing Application Programming Interface (API) to perform the identification of the vehicle's license plate; (ii) use a solar board with a Lithium battery to maintain the electric autonomy of the equipment; (iii) use a service that returns the status and information of the vehicles; (iv) make available a notification mechanism of identified irregular vehicles; (v) make available the occurrences and information of irregular vehicles that circulated through the region through a web interface.

In view of the above, this paper presents the development of an embedded prototype to identify irregular vehicles without the need for human intervention and to serve as an aid to collaborative security based on the IoT framework. The device can be used by anyone and anywhere there is an Internet connection, to check the status of the vehicle or to send notifications about any suspicious driving. Notifications can be sent either to one person or to a group, serving communities that have a community safety system. This pilot project stands out for its low cost and high performance in terms of low computational effort relative to deep learning-based methods.

The main contributions of this paper are:

- The first contribution is related to the proposal of an IoT system that has a low implementation cost and can be used to improve security through an irregular vehicle verification platform.
- The second contribution is related to the use of processing techniques to improve the capability in computer vision-based character identification, which can be employed in other fields to improve image-based pattern classification.
- The results of the application of the proposed model show that it is superior to standard OCR algorithms and has an acceptable response time for a field application

## 2. Related Works

IoT has created a number of new possibilities for improving security [6]. IoT devices can be used to monitor and control access to physical spaces, alert users to potential threats and provide remote access to home and business systems. By connecting physical objects to the Internet, IoT can improve security and reduce the risk of unauthorized access, burglary and other safety risks [7].

Connected door locks and surveillance cameras can help to monitor who is entering a building and whether they have permission to do so. Smart thermostats can be configured to alert users when a door or window has been opened and even turn off the heating system if an intruder has entered the building. Similarly, connected smoke and carbon monoxide detectors can alert users to potential fires [8] and connected security systems can be used to remotely monitor and control access to a building [9].

Artificial intelligence (AI) is increasingly being used in embedded systems, especially in IoT connected devices to assist in improving their information processing capability [10]. The application of AI in IoT enables Internet-connected devices to perform actions autonomously without the need for humans [11]. Examples of AI applications in IoT include using machine learning to analyze sensor data to detect abnormalities or using computer vision to identify objects. In addition, AI can also be used to provide recommendations for actions based on past trends [12].

The techniques of computer vision are increasingly being explored with models of artificial intelligence for the extraction of features that assist in the processing of information [13]. These are issues that might make the problem rather complex, such as the interpretation of two or more characters joined as a single character [14], the interpretation of a point as noise and vice versa, the identification of characters that do not actually exist and the non-interpretation of characters [15].

These difficulties usually occur in situations where characters are connected to drawings. All these challenges make the problem more difficult and advanced models with pre-processing techniques are needed to solve this task. Additionally, artificial intelligence techniques based on convolutional neural networks (CNNs) [16] are increasingly being applied for computer vision analysis [17].

CNNs can be applied to IoT applications to enable the analysis of large amounts of data from connected devices [18]. CNNs can be used to recognize and classify objects in camera images and to detect anomalies [19]. In this paper, the adverse conditions are related to the circulation of irregular vehicles. Deep learning models in particular are increasingly being used for fault detection [20] and power generation forecasting [21], among other prediction applications [22]. IoT, embedded systems [23] and artificial intelligence [24] applications are increasingly being applied for various solutions and are promising for the work at hand.

One of the challenges in developing the interface of this software is to disregard images that are not part of the analysis. For example, when there is some detection of tree movement, all users will be wrongly notified. This means that a simple image classification would not be enough to improve security as in an embedded monitoring system. Embedded systems for IoT [25] are increasingly being used to improve condition monitoring through image analysis, since their application has low cost and high efficiency, making them a promising solution in several fields.

## 3. Applied Framework

This section presents the implementation of an embedded IoT system with the purpose of displaying information pertinent to vehicles that have some irregularity with the documents. This section also describes the specification and architecture used (shown in Figure 1) during the prototype development. To perform the computer vision (CV) analysis, an Intel Core I5-7400, with 20 GB of Random-Access Memory and a GeForce GTX 1050Ti NVIDIA video card was used. CV processing was mainly based on the OpenCV and CV2 libraries using Python 3.9. The embedded system was based on Raspberry Pi 3 hardware.

The first part is the firmware, where a picture is taken when detecting motion. The firmware then processes the image taken via the OpenALPR API [26]. If a card is identified, the geographic coordinates are obtained via GPS and the information is sent to a web service. The web service in turn checks that the data received have not been altered via checksum verification. If the data are genuine, they enter into the database so that the system can query the paid Union Solutions system to see if the vehicle has any irregularities in its documentation.

The main idea in the proposed system is, if there is any irregularity, the system will send an e-mail to a group of people containing a link to access the occurrence. For the user to view the detailed information of the occurrence he must be authenticated in the system. Once authenticated, the user can view the following occurrence information: a point map showing where the device was when identifying the vehicle, license plate, model, color, city, state, passenger capacity, restrictions and the date and time of the event.

In the firmware, the Raspberry Pi 3 was used to build the device with the goal of recognizing the license plate as well as obtaining the geographic coordinates. Internally four threads were created so that the tasks can be executed in parallel. The first thread is responsible for taking a picture. The second thread is responsible for processing the image in order to extract the characters from the image. The third thread is responsible for obtaining the location data. The fourth thread is responsible for sending the information to the web service. The SQLite relational database was used for data persistence and the Python programming language for development.

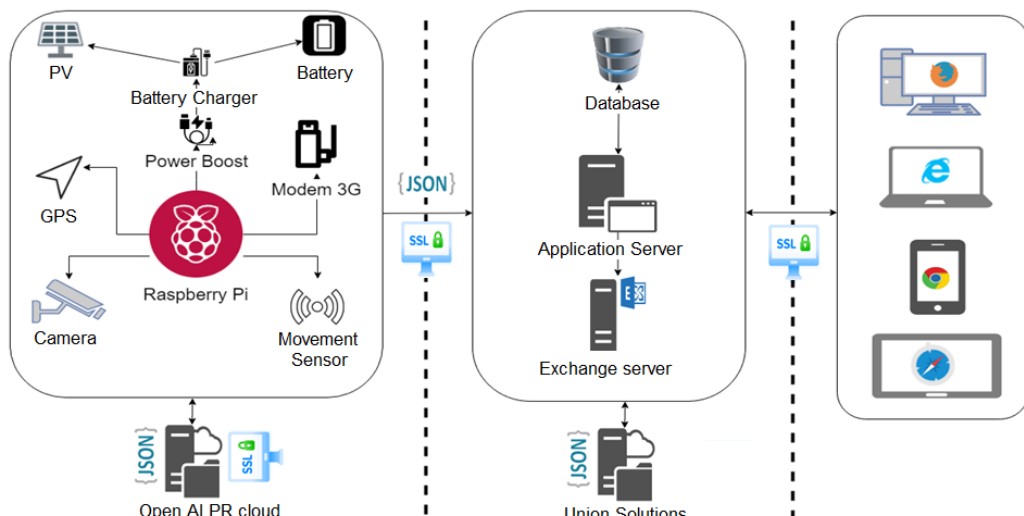

**Figure 1.** Application architecture.

The thread responsible for taking the picture has a main loop that waits for the detection of a movement. Once a movement is detected, a picture is taken. After this, the photo is saved on the firmware's SD card and a record is created in the PICTURE table containing the information about the image name, date and time of the event. The camera used was Raspberry Pi v2 8MP and the motion sensor used was DYP-ME003 manufactured by ElecFreaks. The distance, as well as the delay time, are adjustable on the motion sensor.

The thread responsible for image processing has a loop in which the data from the PICTURE table is read and processed using the OpenALPR API. Image processing is done locally. When performing local processing, the call is made as follows: $alpr - cbr$ $<imagepath>$. If no license plate is recognized, the return is *"No license plates found"*, whereas if license plates are identified you get a list of possible license plates identified and their confidence percentages. Especially within this embedded system, a character detection algorithm was used, as will be explained in the next section.

### 3.1. Optical Character Recognition

Optical Character Recognition (OCR) is a technique that enables automatic character identification [27]. OCR typically consists of optical scanning, localization, segmentation, pre-processing, feature extraction, classification and post-processing [28]. Optical scanning takes place through the process of scanning the image. Typically when performing OCR, one converts the multilevel image into a black-and-white image. According to Stefenon et al. [29], the process of converting the image to black and white can be accomplished through thresholding or binarization [30], where it is possible to apply state-of-the-art methods [31].

In this context, a fixed threshold value is used, where gray levels below this threshold are considered black and levels above it are considered white [32]. For a high-contrast document with a uniform background, a fixed threshold can bring reasonable results. However, many documents encountered in practice have a large contrast range. In these cases, sophisticated thresholding methods are required to obtain a satisfactory result [33].

An example of the pre-processing techniques based on computer vision is shown in Figure 2. Some of the pre-processing techniques can assist in the proper identification of the OCR; however, if no prior evaluation of the results is performed, the processing techniques can reduce the quality of the image and thus hinder the identification and classification of the text. In the example shown in Figure 2 the threshold technique with Otsu and Riddler–Calvard was one of the techniques that most highlighted the characters on the license plate of the vehicle under this pre-evaluation.

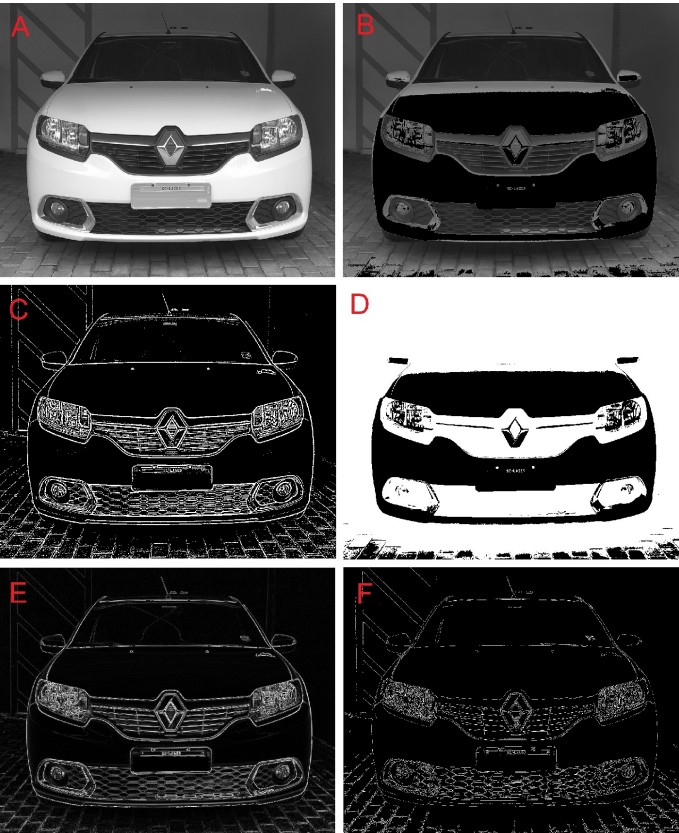

**Figure 2.** Pre-processing images: (**A**) black-and-white converted image; (**B**) binarization with threshold; (**C**) adaptive threshold; (**D**) threshold with Otsu and Riddler–Calvard; (**E**) sobel edge detection; (**F**) canny edge detection.

The binarization process consists of segmenting the characters from the location of the plate. There is an explicit and implicit mode of segmentation. The implicit mode segments the image into several chunks of equal lengths. The explicit mode extracts the characters of the image from the edges of the characters. The main segmentation problems can be divided into four groups: extracting joined or fragmented characters; distinguishing text from noise; interpreting image or graphic as text; interpreting the text as graphic or image. Pre-processing techniques are increasingly necessary for a view of a trend toward the use of deep learning-based models.

### 3.2. OCR Processing

To perform OCR processing for the identification of letters and numbers on license plates [34], initially pre-processing is conducted, resulting in smoothing and noise removal. In addition, analysis and correction of the degree of skew also take place. The degree of skew is the slope value between the baseline of the text and the horizontal line. Filling in smoothing eliminates small breaks and gaps in the scanned characters and reduces the width of the text baseline. Depending on the resolution of the capture medium and the result of the threshold technique applied, characters can become blurred or fragmented. Some of these defects can subsequently cause low recognition rates.

Most of the common techniques for smoothing move a window through the binary image of the character by applying certain rules to the contents of the window. In addition to smoothing, pre-processing usually includes the normalization. Normalization is applied to obtain uniform character size, skew and rotation [35]. To correct rotation, the rotation angle must be found. The Hough transform is usually used to detect distortions in rotated pages and text lines [36]. However, finding the rotation angle of a single symbol is not possible until the symbol has been recognized [37].

The first part is the firmware, where a picture is recorded when the system starts to detect motion. The firmware then processes the image taken via the OpenALPR API. If a card is identified, the geographic coordinates are obtained via GPS and the information is sent to a web service. The web service in turn checks that the data received has not been altered by checksum verification. If the data are genuine, the enter into the database so that the system can query the paid Union Solutions system to see if the vehicle has any irregularities in its documentation.

If there is any irregularity in the vehicle's license plate registration, the system will send an e-mail to a group of people containing a link to access the occurrence. For the user to view the detailed information of the occurrence, he must be authenticated in the system. Once authenticated, the user can view the following occurrence information: a point map showing where the device was when identifying the vehicle, license plate, model, color, city, state, passenger capacity, restrictions and the date and time of the event. In the next sections, the structure and implementation of the modules in the firmware, web service, query, user notification and interface will be detailed.

Processing using OpenALPR occurs through the following steps: detection, binarization, analysis of possible characters, plate edge location, perspective transformation, character segmentation, OCR and post-processing. Figure 3 shows the macro flow of the processing performed via OpenALPR.

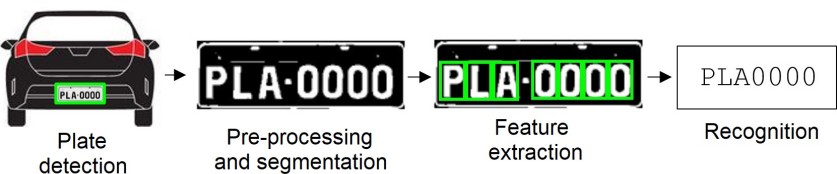

**Figure 3.** OpenALPR macro flow diagram.

The first step uses the Local Binary Pattern (LBP) algorithm, commonly used for face detection, to locate possible plates across regions (x, y, height and width). The binarization step and subsequent ones occur multiple times for each plate region. The binarization step converts the plaque region to black and white. Multiple binary images are used to provide the best possible chance of finding all the characters. The third step, concerning character localization, will try to find character size regions in the plate region. First, points of similar height and width that are in a straight line will be found—this analysis is performed several times on the region in search of small characters and large characters. If nothing is found, the region is discarded and no further processing takes place. If potential characters are found through the dot connection, the region of the possible character will be saved and further processing will take place.

The fourth step consists of locating the edges of the plate, this occurs through the Hough transform to detect vertical and horizontal lines. The fifth step refers to the perspective transform that occurs from the edges of the plate with the purpose of leaving the plate image without rotation and tilt. The character segmentation phase will try to isolate all the characters that make up the image of the license plate. In this procedure, a vertical histogram is used to find gaps in the license plate characters. This phase will also clean up the character boxes by removing small spots and disqualifying smaller character regions. This phase will also attempt to remove regions with "false edges" so that the edge of the plate is not misclassified as a digit or a letter. The seventh step, pertinent to OCR, will be applied with the purpose of extracting each character from the image and classifying it as a digit or letter. The last step, post-processing, determines the best combinations of possible plates where at the top of the list are the *n* best options.

To determine the best options, the program chooses the possibility with the highest degree of confidence, which has letters in the first three digits and numbers in the last four digits. If no valid card is identified, a query is performed in the cloud using the address: api.openalpr.com, (accessed on 1 June 2021). The request to the OpenALPR cloud

is performed through a rest service, the request requires a secret key, the serialized image and also the country parameter. The communication between the firmware and the cloud takes place via the HTTPS protocol. If a card is identified, it is checked whether the card was processed one minute ago. If the board has been processed less than a minute ago, no record is created in the PLATE table.

If the record has not been processed after one minute, a record is created in the PLATE table, along with the date and time of the event. A view was also created with the objective of grouping the data from the PLATE, GPS_COORDINATE and OCCURRENCE tables, allowing the visualization of the last time the license plate was recognized. At the end of the process, the record is deleted from the PICTURE table and also from the SD card. In this way, the same image is not processed more than once. To compile OpenALPR on the Raspberry Pi, it is necessary to have OpenCV installed as well as Tesseract.

The thread responsible for obtaining the geographic data first reads the data present in the PLATE table. After reading the records, the geolocation data are obtained from the Ublox Neo-6m Gy-gps6mv2 GPS and saved in the GPS_COORDINATE table. If the data were successfully fetched, the previously read record is deleted in the PLATE table. While the GPS data are not retrieved, the record is still present in the PLATE table. The thread responsible for sending the information reads the information from the GPS_COORDINATE table and sends it to the web service in JSON format with the information of the card, latitude, longitude, device, checksum, date and time of the event. The use of the checksum guarantees the integrity of the data being transmitted to the web service. If the request to the web service returns the HTTP code 200, a new record will be created in the OCCURRENCE table and deleted in the GPS_COORDINATE table. The communication between the firmware and the web service occurs through the HTTPS protocol.

The firmware has two power sources; the first is the use of the 5.1 V supply that is connected to the mains and the second is the use of the following components: 6 V solar panel, Lithium 3.7 V 2500 mAh battery, battery charger and a 5 V Power Boost with USB output. In this way, the firmware is powered in the presence of light through the solar panel, while in the absence of light, the firmware is powered through the battery. According to the Raspberry Pi documentation, the device should be powered with 5 V and a 2.5 A (2500 mA) power supply is recommended to provide sufficient power.

The firmware can connect to the Internet via wi-fi, cable or 3G modem. Connections through wi-fi and cable are native to the Raspberry Pi. To establish a connection through the 3G modem, the Sakis3g (Trixarian, version 1.0, South Africa, available at: https://github.com/Trixarian/sakis3g-source, (accessed on 1 June 2021)) was used along with the ZTE MF626 3G modem. It was also necessary to install the software *usb_modeswitch* so that the 3G modem is not recognized as a pendrive, but as a modem. A Shell script routine was created with the purpose of checking if the firmware is connected to the Internet; if it is not, a connection to the 3G modem will automatically be established. The connection routine is called every minute; all runs are logged in order to view the information in case an error occurs. The information regarding the APN has been inserted in the following configuration files: */etc/wvdial.conf* and */etc/sakis3g.conf*.

### 3.3. Web Service

The web service developed has the purpose of receiving the data processed by the firmware and entering them into the database. The purpose of the server is to check if the license plate has any irregularities; if there are any, the system will notify a group of people through an e-mail. The site and web service are hosted at Integrator, the main servers are Glassfish application 4.1.2, MySQL database 5.6.44 and E-Mail. The programming language used was Java. The EJB component of JEE was used as well as JPA for data persistence. The Jersey framework was used for the web service implementation.

The firmware makes a request to the web service: https://veiculosirregulares.com.br/vigui/rest/geoPlate, (accessed on 1 June 2021). When the web service receives the information, it first checks if the firmware ID is registered and if it is active. If the firmware is active,

the checksum received is verified. The checksum is calculated using the *HmacSHA*256 algorithm and uses the board information, timestamp, latitude, longitude and the firmware ID. If the checksum is invalid, the record is simply discarded; otherwise, it is checked that the card is valid, i.e., that the first three digits are letters and the next four are numbers. If the card is valid, the record is saved in the database table *row_data* and the *PlateCheckProcessor* class is notified to proceed with the process.

In the license plate verification process, the Observer design pattern was used. This way, as soon as the web service receives information, the verification is invoked. The vehicle documentation verification method consists of the use of the ExecutorService object that enables more executions of the same thread in an asynchronous way. Thus, as the data from the web service are inserted into the system they are processed in parallel. The PlateCheckProcessor thread makes a selection in the *row_data* table and updates the read record column to true. This way, other threads will not process the same record. Once the record is updated, it starts querying the vehicles that are in the database. If the vehicle is already registered and has not been updated for less than a day, the data already in the base will be used to find out if there is an irregularity. If the vehicle is not in the database or is out of date, a request is made to the Union Solutions paid system and the data obtained are saved in the vehicle table.

The vehicle verification will be described later. If an irregularity is found in the vehicle's documentation, the system performs a selection in the *device_group_person_view* in order to notify the group of people by e-mail. To know which group of people should receive an e-mail, a selection is performed in the *device_group_person_view*. The view *device_group_person_view* brings up the information regarding the full name of the people as well as the language that should be used when sending the e-mail. The e-mail sent contains the license plate information of the identified irregular vehicle as well as the link to access the system to view more detailed information, as shown in Figure 4. When an irregularity is found, a record is created in the occurrence table.

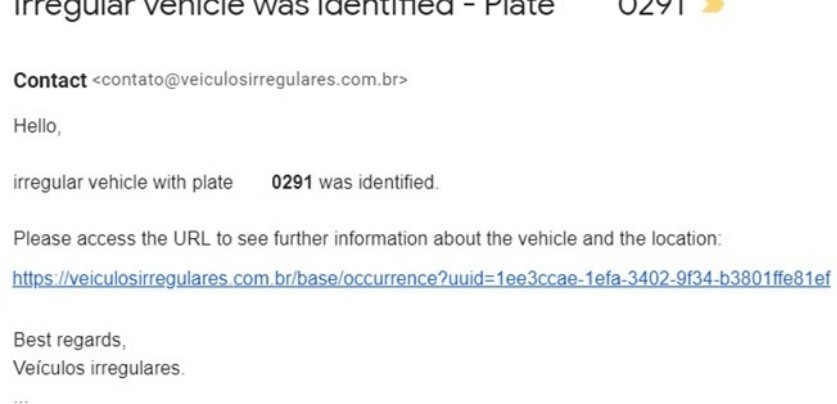

**Figure 4.** Example of an e-mail sent when identifying an irregular vehicle.

The verification process in the Union Solutions paid system consists of a Simple Object Access Protocol (SOAP) request with the license plate number, customer name, login, password and response format. The response format can be XML or JSON. The query return in the Union Solutions system provides several data, such as brand, model, city, federative unit, color, passenger capacity, restriction 1, restriction 2, restriction 3 and restriction 4. The vehicle has a restriction when the return value is different from "NO RESTRICTION".

### 3.4. Interface

The user interface is responsive. Bootstrap, Java Server Faces (JSF), Prime Faces, JavaScript, CSS and HTML were used in its development. For the creation of the site, two free templates from Start Bootstrap were used: Agency and SB Admin. Besides the

responsive interface, the language can be English or Portuguese. For each page, there is a Bean class that fetches and sends the information from the database and shows it to the user. The pages created will be described later.

The following pages are available for navigation: home, login, initial, occurrence search, occurrence, user search, user creation, user change, group search, group creation, group change and password change. The system has user types: administrator and user. The administrator has access to all pages, while the user has access only to the home page, login, initial page, occurrence search, occurrence and password change. The home page contains information about the developer, the purpose of the system and contact information. The login page has the purpose of the system to require authentication through the user's e-mail and password. The home page contains information about the number of irregular vehicles identified in the last 24 h.

To ensure that the system will not be used for other purposes, only registered users will have access, who are responsible for the security of the neighborhood and it is not possible for external users to have information about the information of the system.

The occurrence search page enables the search by license plate, model, make, color and date of the event. The occurrence page contains pertinent information regarding location, license plate, model, color, city, state, passenger capacity, date and time of the event and restrictions. The user search page allows you to search by users' first name, last name and e-mail address. The user change page allows you to change the first name, last name, permission level and language. The create user page allows you to enter first name, last name, e-mail address, permission level and language. When registering a user, the person will receive an e-mail requesting a password change at the first login.

The search, create and change user pages can only be viewed by administrators. The group search page allows you to search by group name only. The group change page allows you to change the group name, and insert and remove users in the group. The create a group page asks you to enter the group's name and participants. The group search, create and change pages can be viewed only by administrators. The password change page allows users and administrators to change their own passwords.

The main page of the system is to view detailed information about the occurrence. When an irregular vehicle is identified, an e-mail is sent to the user group containing a deep link. So, if the user opens the link received he can see all the detailed information. If the user opens the link but is not logged in, he will have to log in. The inactivity time for a session is one hour. After one hour of inactivity, the user must log into the system again.

For user session control, a Filter object was created, whose name is *SessionFilter*. Using a Filter, every request made by the user is checked. If the user tries to open an invalid deep link, he will receive a message saying that he does not have access to the page. The link sent via e-mail contains the *uuid* information. The *uuid* is an *id* that is generated in the database when inserting a new occurrence.

In summary, the method proposed in this paper was to use IoT to capture images, which are then processed using CV in a developed system that has an online platform. In this system, pre-processing techniques are employed to improve OCR performance, only those that meet a confidence criterion are presented. In Figure 5, a comparative result of an input image and the model output after all intermediate steps is presented. As can be seen in this example, OCR returns some false positives when detecting the character "0", which shows the difficulty in performing an accurate detection.

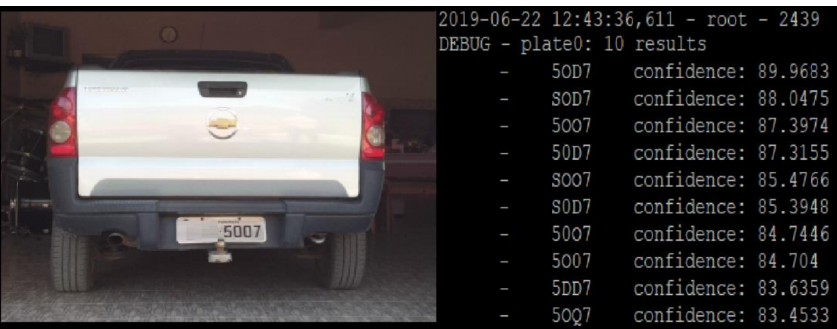

**Figure 5.** Final result compared to the input image in a preliminary test.

## 4. Results and Discussions

This section presents the results obtained in the validation process of the prototype for identifying irregular vehicles. The information regarding the choice of algorithm, the percentage of assertiveness of the license plate recognition, the minimum and maximum distance between the device and the vehicle where the recognition is possible, device angulation and results of the tests with different forms of connectivity are presented here.

Initially, the assertiveness of license plate recognition was validated. Some tests were performed using the following tools: Tesseract, OpenCV with Tesseract and OpenALPR. All the tools used recognized the license plates in high-resolution images where the distance between the vehicle and the camera was short (about one meter). However, only the use of the OpenALPR library was able to recognize the license plates in the low-resolution images captured through the Raspberry Pi camera, as shown in Table 1.

In this analysis, the models based on Tesseract and OpenCV with Tesseract do not satisfy the confidentiality criterion (CC). For this reason, only the OpenALPR was considered. The assertiveness level of the license plate recognition using OpenALPR reached 81.48% out of 108 images with several vehicles positioned at different distances and angles. It was noted that local processing in the firmware is efficient only for stationary vehicles that are within a short distance.

**Table 1.** Performance of the methods.

| Technique | Processing Location | Use of CNN | Percent Assertiv. of Long Distance Images |
|---|---|---|---|
| Tesseract | Local | No | Did not satisfy the CC |
| OpenCV and Tesseract | Local | No | Did not satisfy the CC |
| OpenALPR | Local | Yes | 81.48% |

Next, some tests were performed taking into account the camera positioning. From these, it was identified that the minimum distance for vehicle recognition is 1.15 m, as shown in Figure 6C. The maximum distance for recognition is 7.7 m, as illustrated in Figure 6D. Tests were also performed considering the angulation and it was concluded that the opening degree for capturing the photo is between 55° and 130°, as shown in Figures 6A and 6B, respectively.

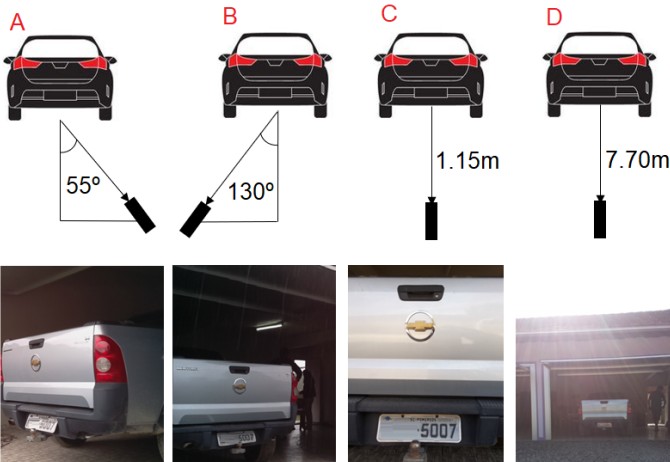

**Figure 6.** Camera positioning: (**A**) minimum angulation; (**B**) maximum angulation; (**C**) minimum distance; (**D**) maximum distance.

From the tests, it was noticed that the aspects related to lighting and temperature are fundamental for image recognition; the image processing can be performed between 8 a.m. and 5 p.m., after this period of time, due to the darkening and increase of the humidity level, the vehicle's license plate is no longer recognized without the aid of complementary lighting. As the day became darker, it was necessary to dry the camera lens; otherwise, the lens would blur making it impossible to read the characters correctly.

The use of the motion sensor was found to be satisfactory. The tests were performed only on a 40 km/h road. In cases where vehicles were traveling below 40 km/h, more pictures were captured. In situations where vehicles were traveling at highway speed, only one photograph was captured. In cases where the vehicles were traveling at a relatively fast speed (approximately 70 km/h), the image was not clear, making it impossible to recognize the license plate. Some tests were performed with the motion sensor fixed next to the Raspberry Pi 5 MP camera along with two other cameras, one being from the iPhone 7 Plus and the other from the Logitech C922 PRO HD webcam, as shown in Figure 7.

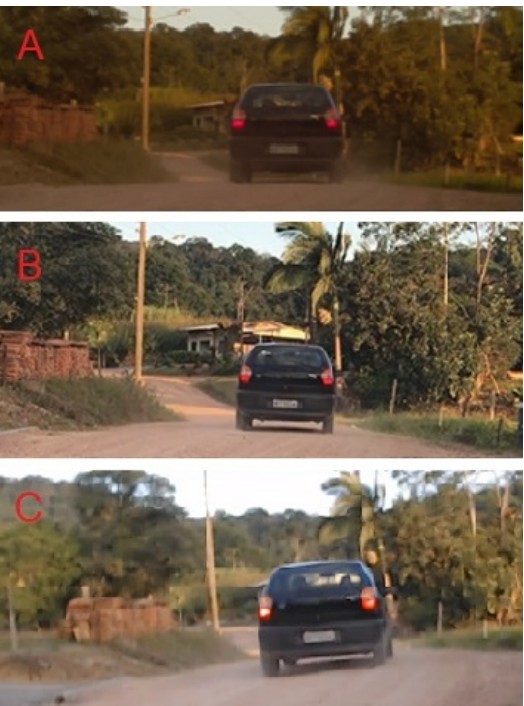

**Figure 7.** Camera: (**A**) RPI (5 MP); (**B**) iPhone 7 Plus; (**C**) Webcam.

It was noticed that in none of the captured images was the license plate recognized through the OpenALPR library due to the long distance and low quality of the Raspberry Pi 5 MP camera. The license plate was only recognized in cases where the vehicle was stationary. To solve the problems of distance and image quality, the fixation of the motion sensor was changed to be below the camera and the Raspberry Pi 5 MP camera was replaced by the Raspberry Pi 8 MP.

Another test performed was in relation to the means of communication (3G, wi-fi and cable). The use of the 3G modem proved to be unfeasible due to the low Internet speed. As in most cases the image processing takes place in the cloud, the waiting time for the image transfer is approximately 76% longer compared to the wi-fi transfer, as shown in Table 2.

**Table 2.** Required response time.

| Connection | Download Speed | Upload Speed | Image Size 800 × 600 | Image Size 1024 × 768 |
|---|---|---|---|---|
| 3G | 0.23 Mbps | 0.12 Mbps | 51 s | 119 s |
| Wi-fi | 2.4 Mbps | 0.7 Mbps | 12 s | 28 s |
| Network cable | 2.5 Mbps | 0.7 Mbps | 9 s | 23 s |

Via testing, it was also found that the average time from receiving the notification including the image processing is 60 s. No tests were performed with an image that has more than one vehicle. No battery life tests were performed either. The tests were performed on a road with low traffic flow, so large-scale tests were not performed and some problems may occur such as firmware overheating, insufficient power supply in the long run, an increase in the delay of the license plate query and notification sending. It was observed that when running the application developed on the Raspberry, there was a significant increase in temperature to 13 ℃ in 11 min.

In addition to detecting irregular vehicles, the embedded system can improve people's safety by monitoring illegal acts, offering a number of features that help detect and prevent criminal actions. Cameras can be installed in strategic locations and can provide valuable information about suspicious activity and help identify possible criminals. Machine learning algorithms and data analysis can be used to detect suspicious behavior patterns and identify possible criminal activity. Face recognition can be used to identify known individuals with criminal backgrounds or who are on watch lists.

A well-designed and implemented embedded system can help improve people's safety by monitoring and preventing illegal acts. However, it is important to remember that such systems can also be used to monitor citizens' activities inappropriately, so it is important to ensure that privacy and data protection policies are respected.

## 5. Conclusions

The development of this article along with the construction of the firmware allowed for a more detailed look at the creation of a mobile embedded system with the goal of using ALPR-based solutions. The prototype has several advantages regarding security, especially in neighborhoods where there is a collaboration among the residents, such as the neighbors' network. However, some problems were identified that hinder the vehicle recognition processing, such as the dust level, the humidity, the temperature, the positioning of the barrel in relation to the street, the attachment mode of the camera with the motion sensor, the state of the characters on the license plate, the unavailability of a fast mobile network, monetary costs and the lack of a firmware cooling system. The architecture used in the server proved to be very robust due to the use of JEE components that are fundamental for scalable systems. The architecture used in the firmware supplies the need to capture pictures and send the pertinent information to the server; in particular, the use of several threads together with a database demonstrated several advantages regarding performance. The use of a 3G modem is not advisable because of the low connection speed.

The use of the Union Solutions paid system proved to be reliable due to the information provided, but problems were noted regarding spelling errors in the description of vehicle restrictions. Thus, a complementary implementation is necessary to show the information correctly to the user. The use of the Union Solutions system can be a problem when dealing with a high number of requests, since each request has a cost of R$2.00. Hosting on the Integrator platform facilitated the configuration of the main servers, besides bringing reliability and avoiding unavailability problems. In general, the use of Raspberry Pi proved to be favorable. However, some aspects such as temperature, installation, cost and energy autonomy should be analyzed with more caution when it comes to having a system working 24 h a day. Depending on the environment where the device is located, it can easily be damaged. The use of OpenALPR proved to be effective for image processing based on the images obtained from the Raspberry Pi camera. In addition, the cloud query in OpenALPR provides information regarding the make, model and color of the vehicle. With this, it is possible to crosscheck the information with Union Solutions in order to verify if the vehicle is cloned.

It can be stated that the cost of the prototype was relatively high (considering a low-cost equipment proposal), adding all the components and not counting freight, the construction cost was R$1630.63. A budget was calculated in some surveillance and monitoring stores and it was concluded that only the value of the camera to identify the plates costs approximately R$8000.00. The prototype presented, besides identifying the license plate, also notifies a group of users informing the geolocation of the equipment. There were also additional costs, such as the acquisition of a cone instead of a beacon barrel and two cameras for the Raspberry Pi. The purchase of the beacon barrel was necessary for the installation of the PV.

The proposed solution has certain limitations, among which we highlight the unfeasibility of vehicle recognition at night and not having a heat sink, thus contributing to the temperature rise due to the high consumption of the processor, the image processing being performed in the cloud in most cases and not locally, Internet connectivity using mobile data, lack of a module to perform real-time monitoring and the use of a button to turn off the device in safe mode without the risk of compromising the integrity of the SD card.

The paper is relevant because it addresses important aspects of the concepts of image processing on Raspberry Pi with an emphasis on public safety, especially in places where there are no high government investments and serves as an escape route for criminals. The prototype can be extended facially to control the entry and exit of vehicles and monitor strategic locations such as parking lots, intersections and speed bumps.

In future works, this project can be extended using state-of-the-art methods such as you only live once (YOLO), which is in its eighth generation and can be implemented in a future project with the specific goal of detecting the position of vehicle license plates. Besides YOLO, other methods could be evaluated to identify the exact position of the vehicle license plate and help the character's location.

**Author Contributions:** Writing—original draft, methodology and formal analysis, L.A.G.; Writing—original draft, M.A.W.; Writing—review and editing, supervision, A.F.H.; Writing—review and editing, S.F.S.; Supervision, A.S. All authors have read and agreed to the published version of the manuscript.

**Funding:** This work was supported by Coordenação de Aperfeiçoamento de Pessoal de Nível Superior.

**Institutional Review Board Statement:** Not applicable.

**Informed Consent Statement:** Not applicable.

**Data Availability Statement:** Not applicable.

**Conflicts of Interest:** The authors declare no conflict of interest.

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
