# Peer review of "OCR Applied for Identification of Vehicles with Irregular Documentation Using IoT"

_electronics, doi:10.3390/electronics12051083_

Round 1

Reviewer 1 Report

This study presents a prototype to identify vehicles in irregular conditions. However, it cannot show some innovation for vehicle identification. Although the authors designed an application architecture, how to develop the firmware and software is unclear. Moreover, there is not any theoretical presentation for the methodologies and the comparison results are unconvincing. More problems are shown as follows:

1. For the application architecture design, what are the improvements and innovations? The authors just repeatedly present the simple processes for the workflow. How to reflect the performance superiority of this designed architecture?

2. In the introduction, the related work is inadequate, and the existing challenges of vehicle identification cannot be summarized.

3.  In Section 2, OCR is an existing technique, it is unnecessary to present too much, about what the application improvement should be enhanced. The authors should give their specific design combining different components and techniques, which is confusing. Moreover, the obtained data just submit to an existing system to handle, what about the contribution of this design? The design of the firmware and system is unclear without any technical presentation.

4. In Section 3, the experiments and results are unconvincing. For example, the Tesseract and OpenCV, and Tesseract give the results of 0, please check the accuracy of the comparison experiments. Moreover, how to reflect the improvement of methodologies and verify the identified performance, more evaluation metrics more details for the experimental design should be given. The citations of Tables 5 and 6 are wrong, and there are no Tables 1 to 3.

5.  In the Conclusion, the authors refer to security, which cannot be shown in the architectural design. How to prove it?

Author Response

Please find the attached response letter.

Reviewer 2 Report

The reviewer thanks the authors for the manuscript, OCR Applied for Identification of Vehicles with Irregular Documentation Using IoT.

1. The paper proposes an embedded system based on a Raspberry having support to GPS, solar panels, communication via 3G modem, wi-fi, camera, and motion sensors. The proposed device manages to address the tough technical concepts pertinent to image processing. The experimental results confirm that this system has excellent performance on the recognition of the license plates, mobility, and autonomy.

2. The introduction part provides a good summary of existing work. Raspberry Pi has been widely used in Internet of things (IoT) systems. More related works on the application of Raspberry Pi should be introduced, such as: doi:10.1007/s00170-020-06394-4; doi:10.3390/electronics11050737.

3. Section 4 presents the experimental results obtained in the test of the prototype for identifying irregular vehicles. The prototype has excellent performance in the validation process. It is recommended that the authors should add more discussion and analysis to this section, to clarify the advantage and novelty related to this embedded system.

Author Response

(The authors gave the same response as above.)

Reviewer 3 Report

This paper outlines a proposal  device that addresses technical concepts pertinent to image processing such as binarization, analysis of possible characters on the plate, plate border location, perspective transformation, character segmentation, OCR, and post-processing.

I advice to include the below comments:

- The introduction is too short  and I advise you to add a small section for related work.

- Include pseudocodes for the mechanism of the system’s work.

- Elaborate section 3 Servidor- line 239

-Put a potential solution to the issue of the system not being able to function in a dark environment.

-Rewrite the conclusion.

Author Response

(The authors gave the same response as above.)

Reviewer 4 Report

The goal of this paper, as exposed by the authors, is to propose a prototype that identifies vehicles in irregular conditions, notifying a group of people, such as a network of neighbors, through a low-cost embedded system based on the IoT.

The authors should discuss the research gap and existing problems in the introduction section as the research motivation. Additional, the authors should summarize their main contributions in this study in bullets in the end of the Introduction section. For each point mentioned in the contribution paragraph, identify which part in the resubmitted manuscript considers that point.

Related work is too short and concise. In order for the article to be accepted for publication, this chapter must be significantly improved both in terms of content size and quality.

What is the frequency at which the CPU works, ISA, RTOS aspects, and what tasks are executed? There are mainly missing aspects related to electronic implementation, considering an article submitted to the Electronics MDPI journal.

Authors should present and validate relevant practical blocks designed by authors such as:

- code section written by authors as personal contribution;

- flowcharts for the software application;

- practical tests on embedded system jitter;

- peripherals used Raspberry Pi;

- Internet stack protocols (TCP/IP) used for IoT;

- practical data on the communication performed by the processor.

For the last step, post-processing, it is only specified that "The last step, post-processing, determine the best combinations of possible plates where at the top of the list are the n best options." In my opinion, more theoretical and practical data can be added on this issue.

Important: Please Quantify the main scientific contributions in the introduction and conclusion sections, compared to other authors' articles. Its reading shows that it is basically a technical article of the use of the IoT with Raspberry, and does not correspond to a new and valuable research in the field.

The reference section is good, citing new and relevant articles in the research area.

Although the paper is interesting, and the authors have put effort into its elaboration, it has some flaws that make it impossible for me to recommend it for publication.

Author Response

(The authors gave the same response as above.)

Round 2

Reviewer 1 Report

All my comments have been addressed.

Reviewer 4 Report

The paper was improved by the revision process.